# MS-YOLOv7:YOLOv7 Based on Multi-Scale for Object Detection on UAV Aerial Photography

**LiangLiang Zhao and MinLing Zhu ***

Computer School, Beijing Information Science and Technology University, Beijing 100101, China;
zhaoliangliang_97@163.com
* Correspondence: zhuminling@bistu.edu.cn

**Abstract:** A multi-scale UAV aerial image object detection model MS-YOLOv7 based on YOLOv7 was proposed to address the issues of a large number of objects and a high proportion of small objects that commonly exist in the Unmanned Aerial Vehicle (UAV) aerial image. The new network is developed with a multiple detection head and a CBAM convolutional attention module to extract features at different scales. To solve the problem of high-density object detection, a YOLOv7 network architecture combined with the Swin Transformer units is proposed, and a new pyramidal pooling module, SPPFS is incorporated into the network. Finally, we incorporate the SoftNMS and the Mish activation function to improve the network's ability to identify overlapping and occlusion objects. Various experiments on the open-source dataset VisDrone2019 reveal that our new model brings a significant performance boost compared to other state-of-the-art (SOTA) models. Compared with the YOLOv7 object detection algorithm of the baseline network, the mAP0.5 of MS-YOLOv7 increased by 6.0%, the mAP0.95 increased by 4.9%. Ablation experiments show that the designed modules can improve detection accuracy and visually display the detection effect in different scenarios. This experiment demonstrates the applicability of the MS-YOLOv7 for UAV aerial photograph object detection.

**Keywords:** UAV; small object detection; YOLOv7; attention mechanism; SPPFS





## 1. Introduction

With the advancement of science and technology, UAVs have become more prevalent in military and civilian applications, and UAV aerials' become a viable task. Object detection in UAV aerial images has consequently become a hot topic in computer vision [1,2]. As the flight height of UAVs is unstable and the image field is expansive, the objects in the UAV aerial images are displayed in a variety of scales, with many small objects, high density, and occlusion between objects, making it hard to capture them accurately. The above problems bring challenges to UAV aerial photography object detection. Determining how to successfully extract object features from UAV aerial images is now essential to resolve the issue [3,4]. It is of great theoretical significance and application value to accurately deal with various complex scenes of UAV object detection.

Currently, techniques for object detection based on deep learning fall primarily into two categories [5]. The first is a two-stage method based on candidate regions, such as the R-CNN series algorithms [6–8], while the second is a one-stage method, such as the SSD [9] and YOLO series algorithms [10–15]. The two-stage method classifies and regresses a series of sparse candidate boxes obtained primarily through heuristic methods and other operations. These two procedures enable the model to achieve the highest level of precision. The one-stage method involves the intensive sampling of different scales and proportions at different locations of the image and then uses a neural network (CNN) to extract image features and classify objects. The one-stage method has high computational efficiency and a fast classification speed, but the unbalanced distribution of positive and negative samples will cause model training and fitting challenges due to the method's uniformly intensive sampling [16].

　　In UAV aerial photography images, there are numerous small, densely distributed objects, and the complex and diverse background environment further complicates object detection, making it difficult to achieve the optimal detection effect. Feng [17] was able to successfully detect small objects in remote sensing images by incorporating motion information into the R-CNN algorithm and adding a smoothing factor to the loss function. However, this method is incapable of detecting multiple-class objects in remote sensing images; it can only detect single-class objects. Li [18] designed the CPN (Category Proposal Network) and F-RPN (Fine-Region Proposal Network) and combined the generated candidate regions with the target number to create the image-adaptive candidate box, which achieves accurate object location and detection. Nonetheless, both the model's size and training complexity are considerable. He [19] developed the remote sensing image object detection model TF-YOLO and proposed a multi-scale object detector based on a deep convolutional neural network, which enabled efficient remote sensing image object detection. However, it did not account for the impact of a complex background on object detection in remote sensing images. Xu [20] incorporated deformable convolution into the network's backbone, extracted multi-scale feature information under different fields of perception, and utilized the network's context information to improve the accuracy of UAV aerial photography small object detection, thereby improving the UAV object detection effect. Pei Wei [21] proposed an improved object detection algorithm for SSD UAV aerial photography images, which effectively fused shallow visual features and deep semantic features of the network via the proposed feature fusion mechanism, thereby alleviating the SSD algorithm's repeated detection issue. Nonetheless, as network depth increases, so does the number of calculations. Despite the fact that the current algorithm has improved the evaluation index, it still has issues with the detection precision of small objects.

　　Due to the problems in the aerial image data set, such as dense targets, a high proportion of small objects, and severe occlusion in some scenes, the object features are not readily apparent; the object detection box features in the data set are quite different from the pre-training data set, making it difficult to achieve optimal results with the general object detection algorithm. Therefore, for the network structure optimization of small objects, the size of the detection head is redesigned to improve the input of characteristic information to the detection head while simultaneously increasing the number and density of anchor frames to improve the network's ability to perceive small objects. The attention mechanism CBAM is incorporated into the backbone network and the feature enhancement network, which suppresses the invalid information in the input features and activates the advantageous features for classification and localization tasks as well as increases the detail and semantic information in the output features. For dense objects, Swin Transformer [22] units are combined with network structure to improve the capture of global information and activate advantageous object location features. In the pooling stage of feature fusion, a new pooling mode, SPPFS, is proposed to address the issue of extracting related repeated features and to enhance the network's information interaction. For box selection when the object is blocked, the index NMS method is applied to prevent false detection and missing detection of the object. In order to penetrate more data into the neural network, a smoother activation function is selected to increase the diversity of feature data and improve the network's ability to detect objects. Based on the above viewpoints, this paper proposes a multi-scale object detection algorithm (MS-YOLOv7). Contributions made by this article are as follows:

(1) We propose a novel network with multiple detection head and introduce the CBAM convolutional attention module to extract features at different scales to improve the detection accuracy of objects at various scales, particularly small objects;

(2) We combine the YOLOv7 network architecture with the Swin Transformer unit and incorporate a new pyramidal pooling module, SPPFS, to solve the problem of high-density object detection;

(3)　　We incorporate the SoftNMS and the Mish() activation function to improve the network's ability to identify overlapping and occlusion objects. Various experiments demonstrate the superiority of our work.

## 2. Related Work

Scale-dependent variation is prevalent in UAV aerial photography. In 2018, Adam [23] proposed some strategies to adapt to the issues in large-scale images based on the YOLOv2 network, including the use of scale transformation, rotation, and other data enhancement operations for various scales and directions. Change the stride from 32 to 16, and modify the network structure and feature fusion for small objects. For large objects, image clipping with overlap is utilized. In 2021, Duan [24] proposed a technique based on a coarse-grained density map. The density estimation network generates a coarse-grained density map, the dense-connected region generates the initial aggregation region, the proportion of objects in the initial aggregation region is estimated, and the initial aggregation region is modified by splitting or enlarging the region. The NMS then returns the conclusive detection result after detecting all clustered areas. In 2021, Zhu [25] proposed TPH-YOLOv5, which adds a prediction head to detect objects of different scales, then uses a Transformer Prediction Head (TPH) combined with a Transformer to replace the original prediction head in order to study the prediction potential with self-attention. This method incorporates some advanced technologies that can effectively improve the performance of aerial image object detection, but the theoretical innovation is limited, and the target occlusion environment is subject to certain constraints. In 2020, Wang [26] used the clustering algorithm to search the region with dense targets, calculated the difficulty value of each clustering region to mine the difficult region and eliminate the simple clustering region, and then used a Gaussian shrinking function to shrink the difficult clustering region in order to reduce the difference between target scales and increase the detection speed. Although this method is straightforward and effective, it cannot be performed in a single pass and must be divided into two stages, from coarse to fine.

Regarding the prevalence of small objects in UAV aerial photography, in 2019, Yang [27] proposed a comprehensive framework for detecting aerial targets. (ClusDet). Scale Estimation Networks (ScaleNet) estimate the scale of target clusters. Cluster areas are identified by a Special Detection Network (DetecNet) following each scale normalization. It can effectively improve the detection performance of dense small objects in aerial remote sensing images, but there are drawbacks, such as the addition of additional networks and the overall structure's complexity. In 2020, Deng [28] proposed an end-to-end Global-Local Adaptive Network. It is composed of three parts: the Global-Local Detection Network (GLDN), the Adaptive Region Selection Algorithm (SARSA), and the Local Super Resolution Network (LSRN). To achieve more precise detection, this method integrates a global local fusion strategy into a network with progressive scale changes. Clipping the original image enables the locally fine detector to detect the object bounding box detected by the globally coarse detector in an auto-adaptive manner. SARSA can dynamically crop dense regions in the input image, whereas LRSN can enlarge the cropped image to provide more detailed information for finer scale extraction and assist the detector in distinguishing between foreground and background. This method is effective, but it still requires predefined superparameters, and it is easy to overlook the target in the image's corner. In 2021, Xu [29] proposed a new AdaZoom network. As a selective magnifying glass with a flexible shape and focal length, it can zoom the focusing area adaptively to detect objects. A reinforcement learning framework is constructed to generate focus areas based on the strategy gradient, and rewards are determined by the object distribution. The region's dimensions and aspect ratio are modified based on the size and distribution of its internal objects. Adaptive multi-scale detection employs variable amplification based on the region's scale. This method employs reinforcement learning, and the effect is unquestionably improved; however, there are some drawbacks, such as setting the reward function, sampling efficiency, and unstable training results. In 2021, Xu [30] proposed a simple and efficient Dot Distance

(DotD), defined as the normalized Euclidean distance between the center points of two boundary frames, which is more appropriate for minimal object detection. This method is currently only utilized in Anchor-based object detectors; it has not yet been investigated in Anchor-free object detectors.

Object occlusion detection is of major concern in UAV aerial photography images. In 2021, Wang [31] proposed the M-CenterNet learning network to improve the detection of minimal objects. After multiple central points were identified, the bias and scale of multiple corresponding objects are computed. In 2022, Li [32] proposed the UAV aerial shot object detection algorithm Acam-YOLO based on the adaptive cooperative attention mechanism and incorporated the Adaptive Co-Attention Module ACAM into the backbone network and feature enhancement network. The input features are sliced in the direction of the channel, and the spatial and channel attention features are extracted, respectively. The synergistic attention weight is then weighted adaptively to improve detection precision. In 2022, Huang [33] proposed a Cascade R-CNN-based algorithm for aerial image detection. On the basis of the original algorithm, superclass detection was added, regression confidence was fused, the loss function was altered, and the ability to detect objects was improved. The primary issue with the aforementioned methods is their limited ability to express the characteristics of small objects against a complex background, as well as their limited detection accuracy.

## 3. Network Structures

The network structure in this paper is shown in Figure 1, which is mainly composed of the backbone, head, and neck. The backbone network continues the network structure of YOLOv7, CBS, and ELLAN modules, which are used for feature extraction to ensure the completeness of feature extraction. The neck uses the multi-scale feature fusion mode of the PAN structure to integrate low-level spatial information and high-level semantic features to preserve more details so as to improve the detection accuracy of small objects. A large-scale detection head is added to the head structure, so as to increase the number and density of the anchor box, and improve the fitting degree of the prediction and object box, thus improving the positioning accuracy. The newly developed SPPFS feature pyramid module is utilized by the backbone to expand the receptive field, enhance the feature information exchange, and improve the detection accuracy of dense objects. A CBAM attention mechanism is added to the neck in order to fully mine information for classification and positioning tasks in input features, which increases the detail richness of output features and enhances the accuracy of network detection. The Swin Transform unit is implanted in front of the head to gather global information, whereas the self-attention mechanism is used to mine the potential of feature expression and activate advantageous features for object location. Finally, the SoftNMS algorithm is used to eliminate redundant prediction boxes to obtain the final results.

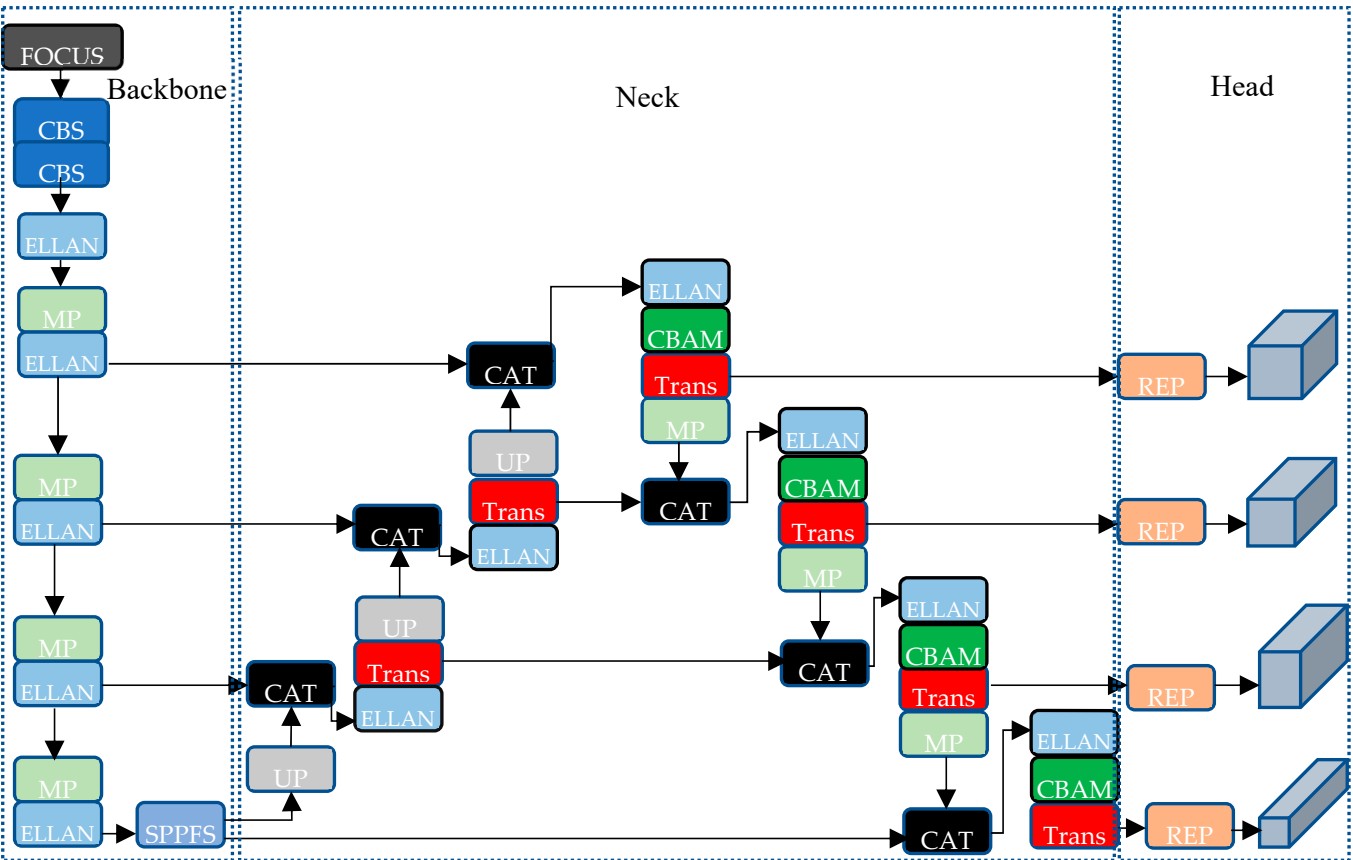

**Figure 1.** Network structure diagram.

## 3.1. Swin Transform

The Swin Transformer structure is contained in the neck of the network as shown in Figure 2. Each module is comprised of two sublayers. The first sublayer utilizes Windows Multi-Head Self-Attention (W-MSA) to divide the feature graph into multiple disjoint regions, with Multi-Head Self-Attention performed only in each region, which reduces the required computation. The second sublayers use a fully connected layer, and the two sublayers are connected by similar residuals. The other module employs Shifted Windows Multi-Head Self-Attention (SW-MSA), which increases the shifting of information across multiple windows. The two modules are typically used in pairs. A Swin Transformer can resolve the issues caused by the CNN's calculation mechanism, such as a bloated model, an excessive number of calculations, and the loss of gradient. A Swin Transformer's input can typically operate directly on pixels to obtain the initial embedding vector, which is closer to how humans perceive the external environment. The multi-head attention model can be utilized to learn pertinent information in distinct subspaces of representation for different tasks and to capture global information. A Swin Transformer has the characteristic of learning long-distance dependence.

In the process of application to the network, the input feature map $(B, H, W, C)$ is initially assigned as a patch $\left(B, \frac{H}{4} * \frac{W}{4}, D\right)$ with a patch size $(4 * 4 * 3)$. While accessing the Swin-Transformer Block module, first slice patches with window_size = 8 to obtain a feature map $\left(B * \frac{H}{M} * \frac{W}{M}, M, M, C\right)$, then calculate the multi-head attention mechanism through W-MSA and SW-MSA in the window, and then restore the original patches $(B, H, W, C)$. Learning features by motion windows not only improves efficiency but also allows neighboring windows to interact with one another, thus creating a cross-window connection between upper and lower layers, thereby achieving a kind of global modeling capability

in a disguised way. It is helpful to solve the problem of object detection in a high-density scene of aerial images.

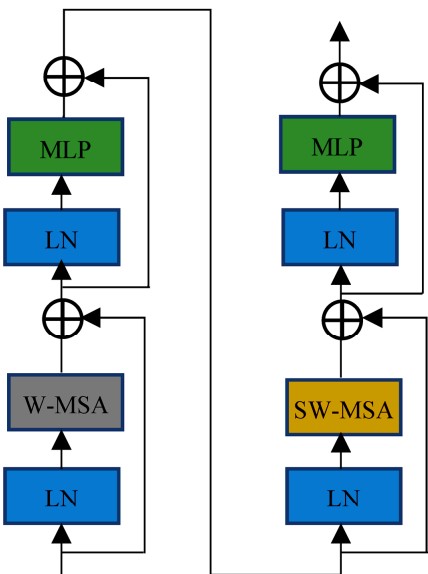

**Figure 2.** Swin Transformer structure diagram.

*3.2. CBAM*

Figure 3 depicts the CBAM module's structure. CBAM is a straightforward, efficient, and lightweight attention module that combines the channel and space attention mechanism modules. It is compatible with most CNN architectures and can be trained end-to-end. Before compressing the space dimension, CBAM determines the attention graph along the channel dimension, which concentrates on the pertinent features of the input image. The attention graph is then derived across the spatial dimension. In contrast to the spatial dimension, the channel dimension is compressed. This module focuses on the location information of the object. The attention graph is multiplied by the input feature graph in order to conduct self-attention feature refinement.

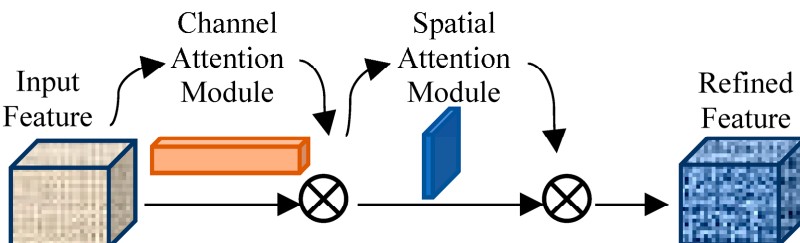

**Figure 3.** CBAM structure diagram.

The feature map $F \in R^{C*H*W}$ is the input of CBAM's attention mechanism module, then through two concurrent MaxPool layers and AvgPool layers to transform size from $C*H*W$ to $C*1*1$, then through the MLP module and ReLU activation function, add element by element to obtain the Channel Attention output. Multiply this output by the original feature map and return it to its original size of $C*H*W$, as indicated by Formula (1). Then through MaxPool layers and AvgPool layers to transform size from $C*H*W$ to $1*H*W$, and the two feature maps are spliced through the Concat operation, then through 7*7 convolution into a 1-channel feature map, and the Spatial Attention output was obtained. Finally, multiply this output by the original feature map and change the size back to $C*H*W$, as indicated by Formula (2). In UAV aerial images, where the object scale changes greatly and intensively, CBAM can be used to extract the attention region

in order to filter out irrelevant information, focus on useful objects and improve model detection performance.

$$M_c(F) = \sigma(MLP(AvgPool(F)) + MLP(MaxPool(F))) \tag{1}$$

$$M_s(F) = \sigma\left(f^{7\times7}([AvgPool(F); MaxPool(F)])\right) \tag{2}$$

### 3.3. SPPFS

Using the design concepts of SPPF and SPPCSPC, we proposed SPPFS for that structure, as shown in Figure 4. The feature map is successively pooled through the different sizes of the pooling kernel, such as $5 \times 5$, $9 \times 9$, and $13 \times 13$, with the step size of 1. Finally, the obtained feature images are concatenated. It is incredible that different pooling kernel sizes have different receptive fields, and multiple feature fusion is conducive to the extraction of dense object features in aerial images. At the same time, feature extraction of different receptive fields is carried out on the same feature map to enhance the interaction of feature information and improve the detection accuracy of object positioning, which is conducive to the detection of objects in high-density scenes of aerial images.

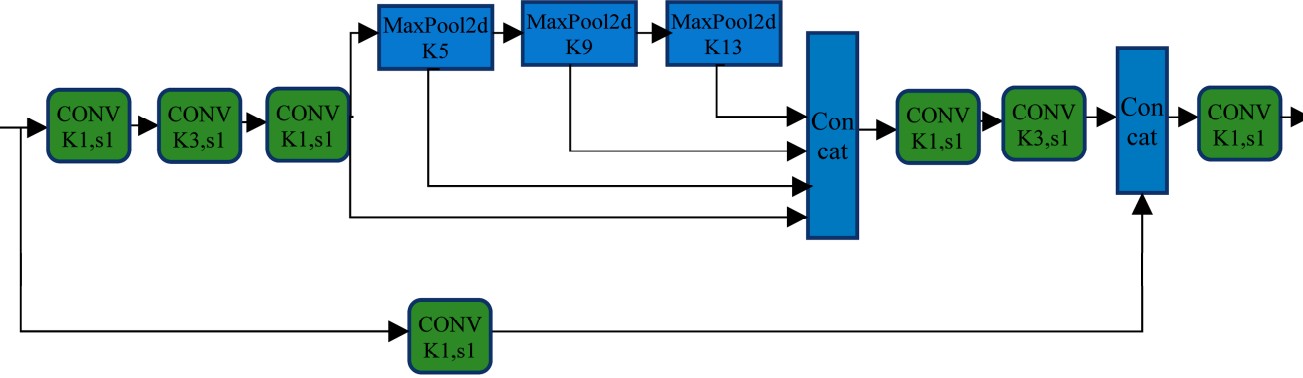

**Figure 4.** SPPFS structure diagram.

### 3.4. Soft-NMS

If the IOU for two objects is substantial, the network generates a substantial IOU for the corresponding Bboxes. The NMS method assumes that the first object has the highest score for a specific Bbox, then records and calculates this object's IOU in comparison to other Bboxes. If the IOU exceeds a predetermined threshold, the associated Bbox is removed, and the Bbox of the object whose IOU is greater than this object's IOU is removed, leaving a single Bbox. When the threshold is too low, it is easy to suppress certain boxes; when the threshold is too high, it is easy to cause false detection, which obscures the inhibition effect.

To resolve the issue with the NMS algorithm, we employ the Soft-NMS algorithm, which performs non-maximal suppression while taking into account the degree of coincidence between the score and the border. Soft-NMS multiplies the obtained IOU by the Gaussian index as a weight, as shown in Formula (3), before reordering and continuing the cycle.

$$s_i = s_i e^{-\frac{IOU(\mathcal{M}, b_i)^2}{\sigma}}, \quad \forall b_i \notin \mathcal{D} \tag{3}$$

$s_i$ is the confidence of the prediction box (Bounding Box), $IOU()$ is the intersection ratio of two frames, $b_i$ is the collection of detected frames, and $\mathcal{D}$ is used to store the final result frame.

Using the CIOU calculation mode in Soft-NMS performs IOU calculations. Figure 5 shows the calculation method for CIOU. CIOU considers the distance between the center points of the actual boxes and the predicted boxes (d in Figure 5) and the diagonal distance between the smallest wrapped box of the two actual boxes (c in Figure 5; the minimum wrapped rectangular box is the dashed line in the Figure). If two boxes do not overlap, the IOU equals 0, causing the failure of backpropagation. CIOU is able to effectively address this issue.

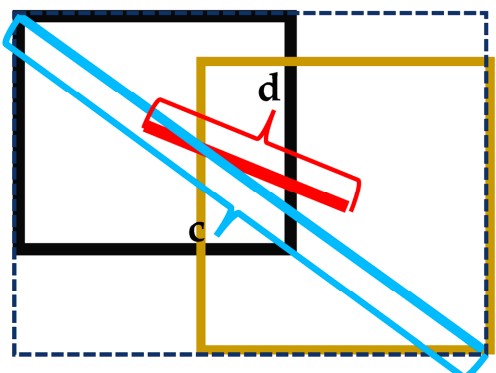

**Figure 5.** Schematic diagram of the CIOU principle.

The calculation method for CIOU is shown in Formula (4) as follows:

$$CIOU = IOU - \frac{\rho^2\left(b, b^{gt}\right)}{c^2} - \alpha v \tag{4}$$

$\rho^2\left(b, b^{gt}\right)$ represents the Euclidean distance of the center point of the prediction box and the real box, respectively, and c represents the diagonal distance of the minimum closure region that can contain both the prediction box and the real box. The calculation methods of $\alpha$ and $v$ are shown in Formulas (5) and (6) as follows:

$$\alpha = \frac{v}{1 - IOU + v} \tag{5}$$

$$v = \frac{4}{\pi^2}\left(arctan\frac{w^{gt}}{h^{gt}} - arctan\frac{w}{h}\right)^2 \tag{6}$$

arctan( ) represents the arctangent function, $w$ and $h$ represent the height and width of the prediction box, and $w^{gt}$ and $h^{gt}$ represent the height and width of the real box.

The pseudo-code of Soft-NMS is as follows:

Input: $\mathcal{B} = \{b_1, \ldots . b_N\}$, $S = \{s_1, \ldots . s_N\}$, $N_t$

    $\mathcal{B}$ is the list of initial detection boxes

    $S$ contains corresponding detection scores

    $N_t$ is NMS threshold

Begin

    $\mathcal{D} \leftarrow \{\}$

    While $\mathcal{B} \neq$ empty do

        $m \leftarrow$ argmax $S$

        $\mathcal{M} \leftarrow b_m$

        $\mathcal{D} \leftarrow \mathcal{D} \cup \mathcal{M}$; $\mathcal{B} \leftarrow \mathcal{B} - \mathcal{M}$

        For $b_i$ in $\mathcal{B}$ do:

            $s_i \leftarrow s_i f(\text{CIOU}(\mathcal{M}, b_i))$

        End

    End

    Return $\mathcal{D}, S$

  end

Soft-NMS solves the problem of missing detection when objects are in close proximity and requires a few additional hyperparameters. It requires only the multiplication of the obtained IOU by the Gaussian index, with no additional memory overhead, and its computational complexity is comparable to that of NMS.

*3.5. Activation Function*

All additional layers of the deep learning model consist of fitting linear functions; although employing a deep neural network for fitting, linear characteristics cannot be avoided, and nonlinear modeling is impossible. In contrast, the output of the linear function undergoes a nonlinear transformation when the output layer passes through the nonlinear activation element. It is possible to complete nonlinear modeling of the input and can also serve as a combination of characteristics.

The Silu() activation function is used for nonlinear activation in YOLOv7. Silu() is shown in Formula (7). It can be seen as a smooth function between the linear function and ReLU function.

$$f(x) = x * sigmoid(x) \qquad (7)$$

The Mish() activation function is calculated as shown in Formula (8).

$$f(x) = x * \tanh(ln(1 + e^x)) \qquad (8)$$

Figure 6 depicts the flowcharts for the Silu() and Mish() functions. The Mish() function is borderless (positive values can reach any height), avoids saturation due to capping, and, in theory, has a better gradient flow for small changes in negative values than ReLU()'s hard zero boundary. Since the Sigmoid() function will have the problem of gradient disappearance, and the problem of gradient disappearance of Tanh() is lighter than Sigmoid(), if the gradient disappears too early, the network convergence rate will be slow. In the positive range, the Mish() function is straighter and smoother than Silu(), and its gradient direction is more consistent than Silu( )'s. The smooth activation function permits more precise and generalized data to enter the neural network, resulting in increased precision and generalization. As a result, Mish() was selected to replace the original Silu() because of its superior ability to express dense occlusion in aerial images.

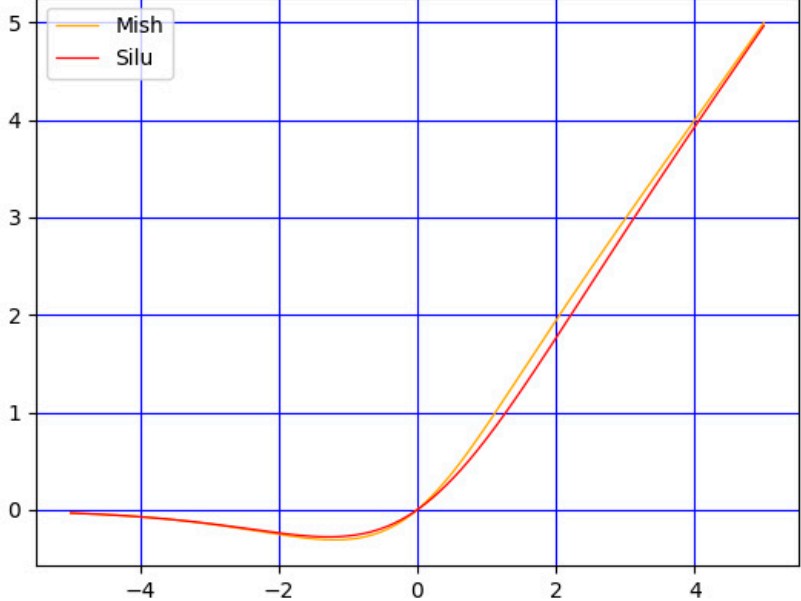

**Figure 6.** Silu() and Mish().

## 4. Experiment

### 4.1. Data Set and Parameter Settings

The VisDrone2019 data set used in this experiment was collected by the AISKYEYE team of the Machine Learning and Data Mining Laboratory at Tianjin University. It is comprised of 6471 images in the training set, 548 images in the verification set, and 3190 images in the test set, all of which contain 10 types of detection objects. There are pedestrians, individuals, bicycles, automobiles, vans, trucks, awning-equipped tricycles, buses, and motor vehicles. Each image in this data set contains a large number of objects, some of which overlap. There are a variety of object sizes and types among the majority of objects; therefore, object detection is somewhat difficult.

The system used in this experiment is Ubuntu18.04, GPU is TeslaP100, memory is 16 GB, CUDA10.02, python3.6, and the deep learning framework used is Pytorch1.8. In the training procedure, the batch size is 16, the Adam algorithm is used to update the gradient, the initial learning rate setting is 0.01, the epoch of a specific amount of training is decreased by a factor of 10, and the momentum is 0.9. The confidence level is 0.5 and $\sigma$ is 0.5 in Soft-NMS.

### 4.2. Comparison of Detection Results of Different Algorithms on VisDrone2019

As shown in Table 1, the detection accuracy of the proposed algorithm is significantly higher than that of other algorithms. $AP_{50}$ is improved by 31.3% relative to Faster R-CNN, by 22.2% relative to YOLOv4, by 12.9% relative to YOLOv8, and by 3.6% relative to ACAM-YOLO in this study. With a large number of small objects, the detection accuracy of pedestrians, people, bicycles, and motorcycles is enhanced. Compared to YOLOv4, the $AP_{50}$ for pedestrians has improved by 38.4%, and the $AP_{50}$ for motorcycles has improved by 31.1%. Regarding the detection of relatively large objects, such as trucks, buses, and automobiles, the precision of detection also possesses obvious benefits. Car detection accuracy is as high as 88.7%, and bus detection accuracy is 70.1%, which is clearly superior to other detection algorithms. The effectiveness of the proposed algorithm for detecting small, dense objects in aerial photography data sets is demonstrated.

**Table 1.** Comparative experiments with different detection algorithms.

| Method | Object Category | | | | | | | | | | $mAP_{0.5}$ |
| --- | --- | --- | --- | --- | --- | --- | --- | --- | --- | --- | --- |
| | Pedestrian | People | Bicycle | Car | Van | Truck | Tri | Awn-Tri | Bus | Motor | |
| Faster R-CNN | 20.9 | 14.8 | 7.3 | 51.0 | 29.7 | 19.5 | 14.0 | 8.8 | 30.5 | 21.2 | 21.8 |
| RetinaNet | 13.0 | 7.9 | 1.4 | 45.5 | 19.9 | 11.5 | 6.3 | 4.2 | 17.8 | 11.8 | 13.9 |
| YOLOv4 | 24.8 | 12.6 | 8.6 | 64.3 | 22.4 | 22.7 | 11.4 | 7.6 | 44.3 | 21.7 | 30.7 |
| CDNet | 35.6 | 19.2 | 13.8 | 55.8 | 42.1 | 38.2 | 33.0 | 25.4 | 49.5 | 29.3 | 34.2 |
| YOLOv3-LITE | 34.5 | 23.4 | 7.9 | 70.8 | 31.3 | 21.9 | 15.3 | 6.2 | 40.9 | 32.7 | 28.5 |
| MSC-CenterNet | 33.7 | 15.2 | 12.1 | 55.2 | 40.5 | 34.1 | 29.2 | 21.6 | 42.2 | 27.5 | 31.1 |
| DMNet | 28.5 | 20.4 | 15.9 | 56.8 | 37.9 | 30.1 | 22.6 | 14.0 | 41.7 | 29.2 | 30.3 |
| HR-Cascade++ | 32.6 | 17.3 | 11.1 | 54.7 | 42.4 | 35.3 | 32.7 | 24.1 | 46.5 | 28.2 | 32.5 |
| DBAI-Net | 36.7 | 12.8 | 14.7 | 47.4 | 38.0 | 41.4 | 23.4 | 16.9 | 31.9 | 16.6 | 28.0 |
| Cascade R-CNN | 22.2 | 14.8 | 7.6 | 54.6 | 31.5 | 21.6 | 14.8 | 8.6 | 34.9 | 21.4 | 23.2 |
| CenterNet | 22.6 | 20.6 | 14.6 | 59.7 | 24.0 | 21.3 | 20.1 | 17.4 | 37.9 | 23.7 | 26.2 |
| MSA-YOLO | 33.4 | 17.3 | 11.2 | 76.8 | 41.5 | 41.4 | 14.8 | 18.4 | 60.9 | 31.0 | 34.7 |
| TPH-YOLOv5 | 29.00 | 16.7 | 15.6 | 68.9 | 49.7 | 45.1 | 27. | 24.7 | 61.8 | 30.9 | 37.3 |
| ACAM-YOLO | 57.6 | 45.9 | 25.7 | 88.5 | 51.9 | 45.5 | 38.1 | 19.9 | 69.1 | 56.3 | 49.5 |
| YOLOv8 | 50.2 | 39.7 | 21.3 | 74.8 | 50.5 | 46.2 | 33.3 | 22.1 | 67.4 | 45.3 | 40.2 |
| Ours | **63.2** | **51.7** | **26.2** | **88.7** | **56.2** | **48.7** | **39.7** | **21.8** | **70.1** | **63.8** | **53.1** |

### 4.3. Ablation Experiment

To determine the effectiveness of aerial image detection by the Small Object Detection head (Small), Swin Transformer module (Sw T), CBAM module (CBAM), SPPPFS module (SPP), Mish() function (Mish), and Soft-NMS mode (Soft), an ablation experiment of the detection network was conducted utilizing the YOLOv7 model. The size of the input image was $640 \times 640$ pixels, the batch size was set to 10, and the training time for each network was 400 epochs. The experimental results are shown in Table 2.

**Table 2.** Ablation experiment.

| Method | $mAP_{0.5}$ | $mAP_{0.95}$ | Parameters |
| --- | --- | --- | --- |
| YOLOv7 | 47.1 | 26.4 | 71.4 M |
| YOLOv7 + Sw T | 49.5 | 27.9 | 77.2 M |
| YOLOv7 + Small | 51.8 | 30.7 | 74.4 M |
| YOLOv7 + Mish | 49.3 | 27.9 | 71.4 M |
| YOLOv7 + SPP | 49.3 | 27.7 | 71.4 M |
| YOLOv7 + CBAM | 49.5 | 28.0 | 62.9 M |
| YOLOv7 + Soft | 49.7 | 28.1 | 71.4 M |
| YOLOv7_ Small _Sw T | 51.5 | 30.6 | 79.7 M |
| YOLOv7_ Small _Sw T_ CBAM | 51.6 | 30.4 | 79.7 M |
| YOLOv7_ Small _Sw T_ CBAM _SPP | 52.1 | 30.9 | 79.7 M |
| YOLOv7_ Small _Sw T_ CBAM _SPP _ Mish | 52.8 | 31.1 | 79.7 M |
| YOLOv7_ Small _Sw T_ CBAM _SPP _ Mish _Soft | **53.1** | **31.3** | 79.7 M |

According to Table 2, the $mAP_{0.5}$ of the baseline network of YOLOv7 under the same conditions is 47.1%, and the optimization of the network structure for $mAP_{0.5}$ and $mAP_{0.95}$ results in a significant improvement. By adding a Swin Transformer unit, $mAP_{0.5}$ has increased by 2.4%, but due to an increase in the feedforward neural network the parameter quantity has increased by 5.8 M. With the addition of the prediction head for small object detection, $mAP_{0.5}$ can be enhanced by 4.7%, effectively capturing the characteristics of small objects and enhancing the ability to detect them. But due to the addition of a small object detection head, the number of parameters was slightly increased by 3 M. Using the new feature pyramid SPPFS module, $mAP_{0.5}$ is enhanced by 2.2%, indicating that a large receptive field is advantageous for object detection. By adding the CBAM module, $mAP_{0.5}$ has increased by 2.4%, and the parameter quantity has decreased by 8.5 M. CBAM efficiently extracts the channel and spatial information of input features, enables the use of effective information in multiple dimensions, improves the performance of the detection network, and reduces the number of model parameters. The choice of activation function enables superior data to enter the neural network, resulting in enhanced precision and $mAP_{0.5}$ is enhanced by 2.2%. The non-maximum suppression method was improved to circumvent the problem of undetected objects resulting from a low object density.

Although some of the above methods increase the number of parameters, the detection accuracy has improved. For the detection of dense aerial data sets containing object information, it is suggested that the use of small-head detection methods has clear advantages. The Swim Transformer is capable of capturing global characteristics and learning long-distance dependencies in an efficient manner. The CMAB attention module can realize channel and spatial dimension information effectively and increase the rate of effective information utilization in input features. The Mish() function has exceptional nonlinear expression capability. The SoftNMS effectively avoids the problem of missed objects.

### 4.4. Visualization

To verify the visual effects of the proposed algorithm in various aerial image detection scenarios, we selected images from the VisDrone2019 test set for detection under high-altitude visual field, complex scene, and object occlusion conditions; the result is demonstrated in Figure 7. Figures 8 and 9 depict a visual comparison with the detection

effect of the YOLOv7 baseline model in the same scene. Figure 10 shows the confusion matrix diagrams for YOLOv7 and this article.

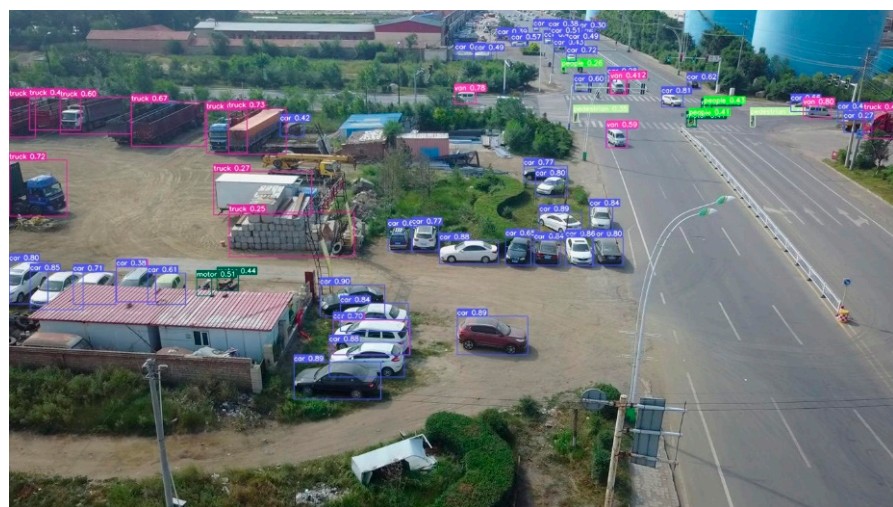

(**a**) **Detection diagram for complex scenarios**

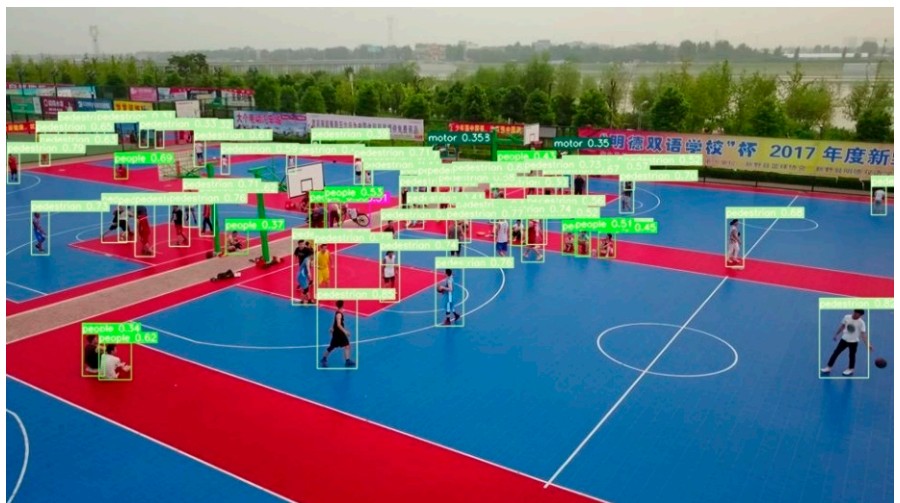

(**b**) **Detection diagram under object occlusion**

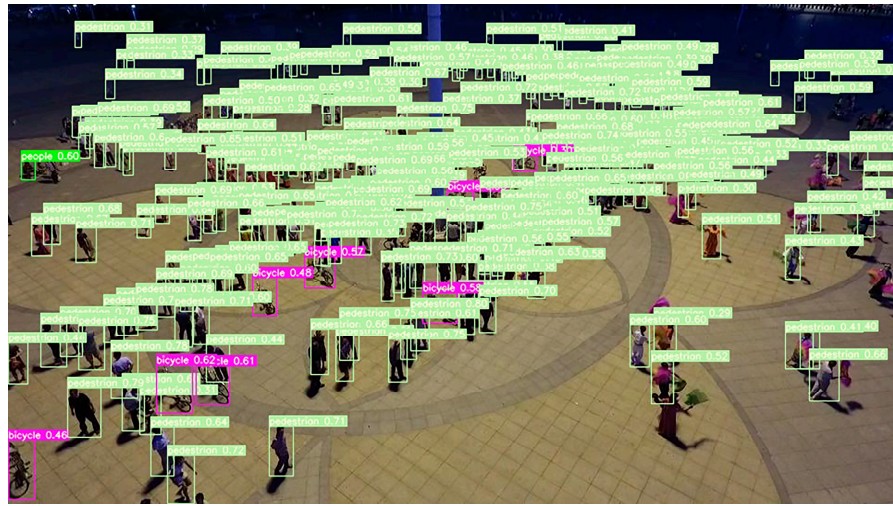

(**c**) **Detection map under light changes**

**Figure 7.** *Cont.*

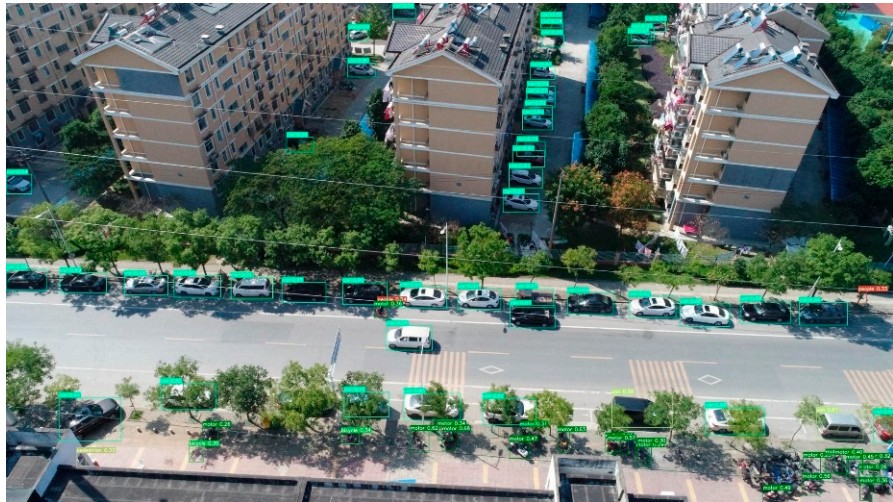

(**d**) Detection map under high visual field

**Figure 7.** Test effect diagrams under different conditions.

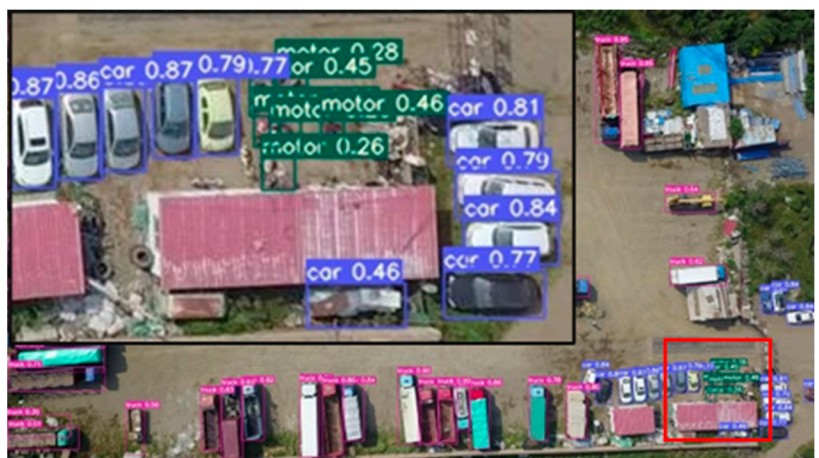

(**a**) Algorithm of this paper

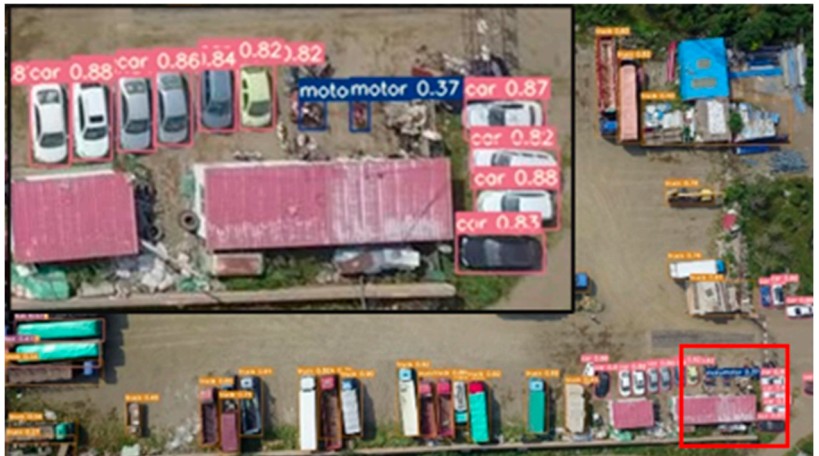

(**b**) Baseline Model (YOLOv7)

**Figure 8.** Comparison of overhead detection effects.

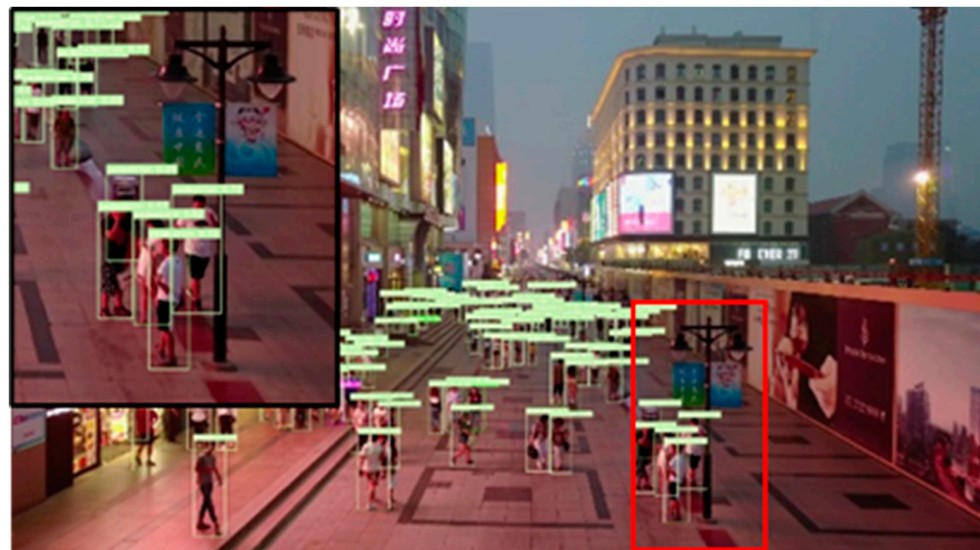

**(a) Algorithm of this paper**

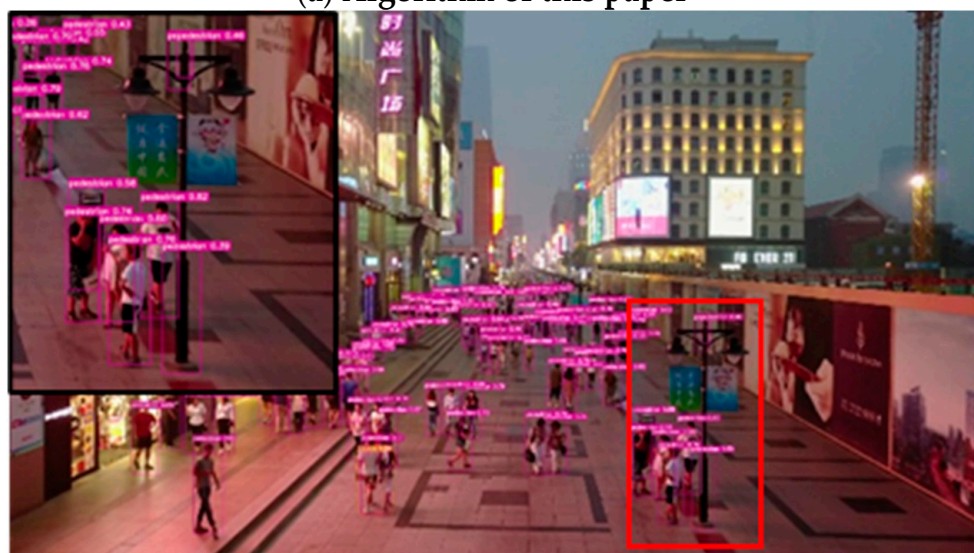

**(b) Baseline Model (YOLOv7)**

**Figure 9.** Comparison of night detection effects.

Figure 7a demonstrates that under conditions of shrub cover, accumulation of building materials, and complex road conditions, the detection effect of vehicles and pedestrians is good, and the motorcycle object that appears in front of the house can also be clearly detected. As depicted in Figure 7b, it is still possible to accurately detect playground pedestrians in heavily shielded environments. As depicted in Figure 7c, shielded pedestrians can be detected in a square with low light and crowded people, and bicycles among pedestrians can also be accurately detected. As shown in Figure 7d, in the environment of high-altitude buildings and roads, it is possible to detect all vehicles in the shade of trees and between buildings. It demonstrates that our method is highly applicable to the detection of aerial UAV images.

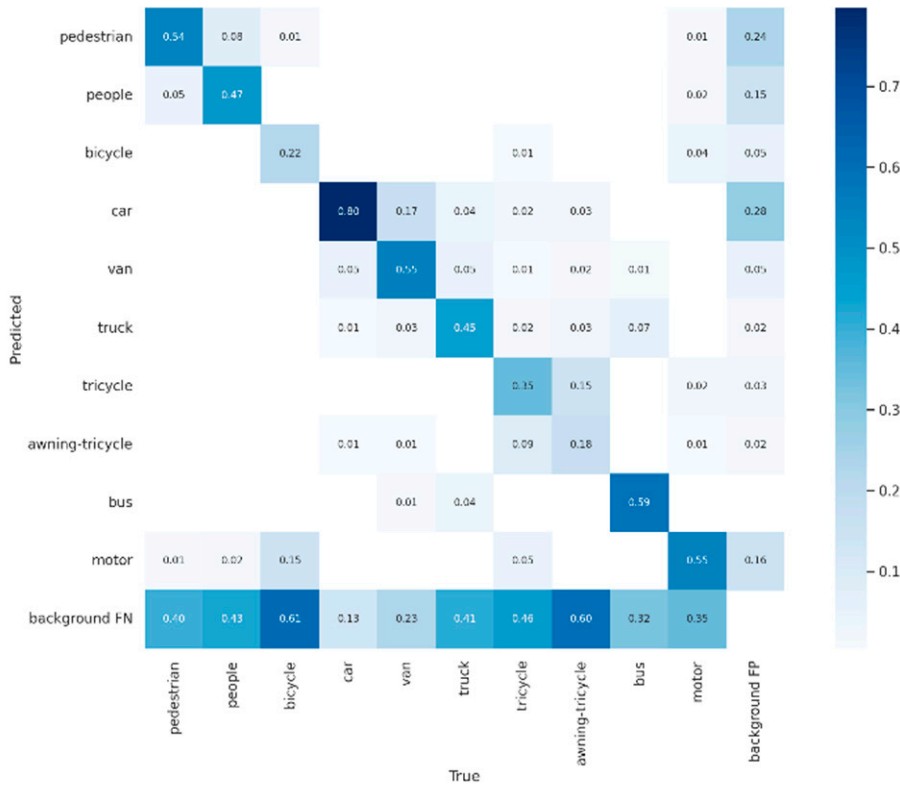

(**a**) **YOLOv7**

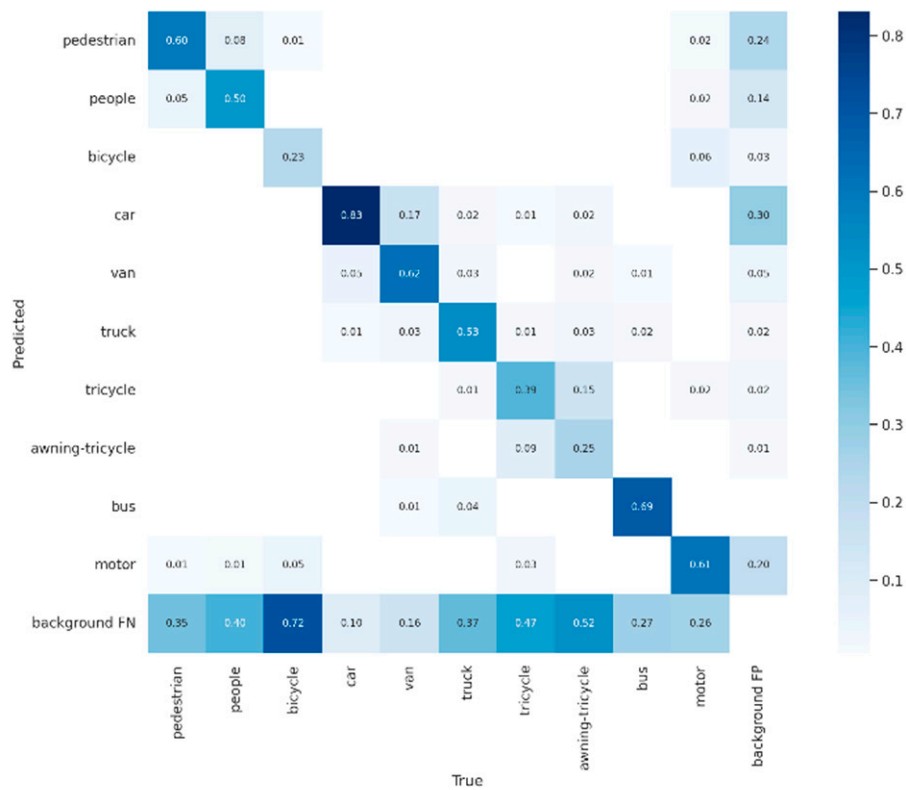

(**b**) **Algorithm of this paper**

**Figure 10.** Confusion matrix diagram.

Figure 8a,b depicts a comparison diagram of vehicle detection on the ground in a scene viewed from a high altitude. In the case of dense small objects, YOLOv7 failed to detect the motorcycle objects adjacent to the vehicle, whereas the method presented in this paper detected every motorcycle small object.

The detection comparison diagram depicts a nighttime pedestrian occlusion scene, as shown in Figure 9a,b. In the case of multiple pedestrians and street lamp occlusion, YOLOv7 incorrectly identifies street lamps as pedestrian objects, whereas the method outlined in this paper correctly identifies pedestrian objects. It demonstrates that our method detects dense occlusions in UAV aerial images with precision.

Figure 10a,b, respectively, show the confusion matrix diagrams for YOL0v7 and this article. Each row of the confusion matrix corresponds to a real class, while each column corresponds to a predicted class. For car objects, the detection precision of the two networks is comparable. For other objects, the proposed method has a significantly higher detection accuracy than YOLOv7. The proposed method has a lower false detection rate than YOLOv7.

## 5. Conclusions

In this paper, a multi-scale UAV aerial image object detection algorithm, MS-YOLOv7, is proposed for the study of the UAV aerial image object detection task. For the network structure optimization of small objects, the size of the detection head is redesigned to improve the input of characteristic information to the detection head, while simultaneously increasing the number and density of anchor frames to improve the network's ability to perceive small objects. The attention mechanism CBAM is incorporated into the backbone network and the feature enhancement network, which suppresses the invalid information in the input features and activates the advantageous features for classification and localization tasks as well as increases the detail and semantic information in the output features. For dense objects, Swin Transformer units are combined with network structure to improve the capture of global information and activate advantageous object location features. The new pyramid feature fusion module SPPFS increases the feature acceptance range and the capacity to represent feature graphs. The Mish() activation function has a more accurate and smoother nonlinear representation. The SoftNMS algorithm employing the probability distribution function as the weight of the attenuating confidence score reduces the possibility that the blocked object frame will be mistakenly deleted and improves the object's recognition capability.

The proposed algorithm has a clear advantage over other algorithms in the publicly available data set VisDrone2019 for the 10 types of objects with the highest $mAP_{0.5}$ values. In comparison to the YOLOv7 baseline model, the $mAP_{0.5}$ and $mAP_{0.95}$ values have significantly increased. The ablation experiment demonstrates that each component of the design has significantly improved the accuracy of detection, and the visualization experiment demonstrates that it has a good detection effect in various scenes. But there is room for improvement in the detection effect of MS-YOLOv7 for aerial images with severe occlusion. This avenue is a key research direction for aerial image-intensive small object detection in the future.

**Author Contributions:** Conceptualization: L.Z. and M.Z., methodology: L.Z. and M.Z., formal analysis: L.Z. and M.Z., investigation: L.Z. and M.Z., data curation: L.Z., writing– original draft preparation: L.Z., writing–review and editing: L.Z. and M.Z. All authors have read and agreed to the published version of the manuscript.

**Funding:** This research was funded from the support of the subject and research project of School of Computer Science, Beijing Information Science and Technology University, and the National Natural Science Foundation of China (No. 31900979).

**Institutional Review Board Statement:** Not applicable.

**Informed Consent Statement:** Not applicable.

**Data Availability Statement:** Data set: https://github.com/VisDrone.

**Acknowledgments:** We thank the company ZSE, a.s., for supporting the open-access publication of this paper.

**Conflicts of Interest:** The authors declare no conflict of interest.

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
