# Peer review of "MS-YOLOv7:YOLOv7 Based on Multi-Scale for Object Detection on UAV Aerial Photography"

_drones, doi:10.3390/drones7030188_

Round 1

Reviewer 1 Report

1. In this paper, "target detection” and "object detection" are both used, please keep consistent with the title.

2. Page4/ Figure 1: Please check that the box, do they have an upper boundary?

3. Page7 Please explain the parameters in the formula (4).

4.  Page8/ Table 1, Page9 / Table 2: Please highlight the best accuracy values in bold.

5.  Page10/ Line 363: Please modify the format of "YOL0v7".

6. It is suggested to conduct a contrast experiment with YOLOv8.

7. The English should be checked for grammar mistakes.

Author Response

  1. In this paper, "target detection” and "object detection" are both used, please keep consistent with the title.

We changed "target detection" to "object detection" in the article.

  1. Page4/ Figure 1: Please check that the box, do they have an upper boundary?

We completed the upper boundary of the Box in Figure 1.

  1. Page7 Please explain the parameters in the formula (4).

We have explained the parameters in the formula. It is shown in red font in P.8/line289.

  1. Page8/ Table 1, Page9 / Table 2: Please highlight the best accuracy values in bold.

We showed the best accuracy values in bold in Table 1 and Table 2.

  1. Page10/ Line 363: Please modify the format of "YOL0v7".

We changed "YOL0v7" to "YOLOv7" in Page12/ Line 406.

  1. It is suggested to conduct a contrast experiment with YOLOv8.

We added a comparison test with YOLOv8 in Table1.

  1. The English should be checked for grammar mistakes.

We improved the English grammar of the article. such as “In this paper, a multi-scale UAV aerial image target detection algorithm based on MS-YOLOv7 is proposed for the study of UAV aerial image target detection task.” to “In this paper, a multi-scale UAV aerial image object detection algorithm MS-YOLOv7 is proposed for the study of UAV aerial image object detection task”.

Reviewer 2 Report

The submission proposes a multi-scale object detection network, named MS-YOLOv7, for the challenges of detecting a large number of targets and a high proportion of small targets on UAV imagery. The MS-YOLOv7 adds several improved modules based on the YOLOv7 network, including the target prediction head, swin transformer attention module, CBAM, a new SPPFS, etc. Authors also evaluate the performance of the proposed framework on publicly VisDrone2019 dataset. However, I have a number of concerns with the manuscript, which are outlined below.

1-   The innovation in contributions needs to be further condensed. Meanwhile, a whole bunch of added modules are more like a stack of modules on the advanced YOLOv7 network. Since quite a few modules are widely used in the field of target detection, the authors should further explain the significance of choosing these modules together.

2-   Sections 3.2 and 3.3, along with Figure 1 provide a rather superficial description of the proposed architecture. It would be good for the authors to provide info on channel dimensions and describe exact operations in the figures.

3-   The parameter size of the model is also unclear. The authors should include this in their comparison table with the other methods or baselines.

4-   Figure 4 shows that the features of different sizes after maxpool2d are connected by “concat” operation directly. I think there should be an error.

5-   I don't know how this conclusion was drawn: “The Mish() function ……, in theory, a better gradient flow for small changes in negative values than ReLU()'s hard zero boundary” (in lines 291-294).

6-   The horizontal and vertical coordinates of Figure 6 are not clear.

7-   What does “changes” mean in this sentence “the difficulties posed by large-scale changes” (in line 72)?

8-   The submission repeatedly mentioned that the motivation is to improve the accuracy of small targets, but ignore the comparative experiments on small targets.

Author Response

  1. The innovation in contributions needs to be further condensed. Meanwhile, a whole bunch of added modules are more like a stack of modules on the advanced YOLOv7 network. Since quite a few modules are widely used in the field of target detection, the authors should further explain the significance of choosing these modules together.

In Page2/ Line 72, we introduced the significance of designing different modules for different problems, and in Sections 3, we introduceed different modules in more detail. The text is shown in red.

  1. Sections 3.2 and 3.3, along with Figure 1 provide a rather superficial description of the proposed architecture. It would be good for the authors to provide info on channel dimensions and describe exact operations in the figures.

The network architecture is further explained completely in Page4/ Line 173, and detailed operations are described in Sections 3.1, 3.2 and 3.3. The text is shown in red.

  1. The parameter size of the model is also unclear. The authors should include this in their comparison table with the other methods or baselines.

We increased the number of model parameters in the ablation experiment, as shown in Table 2.

  1. Figure 4 shows that the features of different sizes after maxpool2d are connected by “concat” operation directly. I think there should be an error.

Because the step of the pooling operation is 1, the feature graph size does not change, so the “concat” operation can be performed directly.

  1. I don't know how this conclusion was drawn: “The Mish() function ……, in theory, a better gradient flow for small changes in negative values than ReLU()'s hard zero boundary” (in lines 291-294).

We explained its conclusion in Page8/ Line 326. The text is shown in red.

  1. The horizontal and vertical coordinates of Figure 6 are not clear.

The horizontal and vertical coordinates in Table 6 were clarified.

  1. What does “changes” mean in this sentence “the difficulties posed by large-scale changes” (in line 72)?

The unstable flight height of UAV leads to the change of target scale, which brings difficulties to detection.

  1. The submission repeatedly mentioned that the motivation is to improve the accuracy of small targets, but ignore the comparative experiments on small targets.

Because this paper aims at the target detection of UAV aerial images, the target detected is actually a small target.

Reviewer 3 Report

The logic of the article is clear, the diagrams are properly explained. The experiment is complete, and the experimental results are explained in detail. But there are still some deficiencies.

(1) When introducing the network structures, I noticed that some existing modules and designs have no relevant references. Specific references should be given to some previous achievements.

(2) For the network structure, if possible, you should provide more detailed design description and mathematical analysis. A more in-depth discussion of how they work is not just a simple introduction.

(3) Considering that the loss function is an important part of the network and whether the loss function of the network should be introduced.

Author Response

  1. When introducing the network structures, I noticed that some existing modules and designs have no relevant references. Specific references should be given to some previous achievements.

In section 3, we further explain the network module in detail. The text is shown in red.

  1. For the network structure, if possible, you should provide more detailed design description and mathematical analysis. A more in-depth discussion of how they work is not just a simple introduction.

The network architecture is further explained completely in Page4/ Line 173, and detailed operations are described in Sections 3.1, 3.2 and 3.3. The text is shown in red.

  1. Considering that the loss function is an important part of the network and whether the loss function of the network should be introduced

Because we have used the loss function of YOLOv7 and did not modify it, which could also achieve good detection effect. we didn’t introduce it in the paper.

Reviewer 4 Report

The manuscript entitled “MS-YOLOv7:YOLOv7 Based on Multi-scale for Object Detection on UAV Aerial Photography” has been investigated in detail. The topic addressed in the manuscript is potentially interesting and the manuscript contains some practical meanings, however, there are some issues which should be addressed by the authors:

1)      In the first place, I would encourage the authors to extend the abstract more with the key results. As it is, the abstract is a little thin and does not quite convey the interesting results that follow in the main paper. The "Abstract" section can be made much more impressive by highlighting your contributions. The contribution of the study should be explained simply and clearly.

2)      The readability and presentation of the study should be further improved. The paper suffers from language problems.

3)      The importance of the design carried out in this manuscript can be explained better than other important studies published in this field. I recommend the authors to review other recently developed works.

4)      “Discussion” section should be added in a more highlighting, argumentative way. The author should analysis the reason why the tested results is achieved.

5)      The authors should clearly emphasize the contribution of the study. Please note that the up-to-date of references will contribute to the up-to-date of your manuscript. The studies named "Artificial intelligence-based robust hybrid algorithm design and implementation for real-time detection of plant diseases in agricultural environments; Detection of solder paste defects with an optimization‐based deep learning model using image processing techniques; Recognition of COVID-19 disease from X-ray images by hybrid model consisting of 2D curvelet transform, chaotic salp swarm algorithm and deep learning technique"- can be used to explain the method in the study or to indicate the contribution in the “Introduction” section.

6)      How to set the parameters of proposed method for better performance?

7)      It will be helpful to the readers if some discussions about insight of the main results are added as Remarks.

This study may be proposed for publication if it is addressed in the specified problems.

Author Response

  1. In the first place, I would encourage the authors to extend the abstract more with the key results. As it is, the abstract is a little thin and does not quite convey the interesting results that follow in the main paper. The "Abstract" section can be made much more impressive by highlighting your contributions. The contribution of the study should be explained simply and clearly.

We revised the abstract to highlight our contributions and extended some key experimental results and types of experiments to the abstract. The text is shown in red.

  1. The readability and presentation of the study should be further improved. The paper suffers from language problems.

We have modified the grammar to make the paper more readable.

  1. The importance of the design carried out in this manuscript can be explained better than other important studies published in this field. I recommend the authors to review other recently developed works.

We referred to other articles in the field and explained the importance of our research in the introduction.

  1. “Discussion” section should be added in a more highlighting, argumentative way. The author should analysis the reason why the tested results is achieved.

In the conclusion, we highlight the salient points of this study, analyze the reasons for the results, and further look into the future work. The text is shown in red.

  1. The authors should clearly emphasize the contribution of the study. Please note that the up-to-date of references will contribute to the up-to-date of your manuscript. The studies named "Artificial intelligence-based robust hybrid algorithm design and implementation for real-time detection of plant diseases in agricultural environments; Detection of solder paste defects with an optimization‐based deep learning model using image processing techniques; Recognition of COVID-19 disease from X-ray images by hybrid model consisting of 2D curvelet transform, chaotic salp swarm algorithm and deep learning technique"- can be used to explain the method in the study or to indicate the contribution in the “Introduction” section.

In the introduction, we highlight the contribution of this research and explain the main research methods. The text is shown in red.

  1. How to set the parameters of proposed method for better performance?

The main parameters of the paper are explained in Sections 4.1 Experimental Settings.

  1. It will be helpful to the readers if some discussions about insight of the main results are added as Remarks.

We did a lot of experiments and discussed the results in detail in section 4, so that the reader could understand them more clearly.

Round 2

Reviewer 2 Report

I have no further comments. Good luck to your research.

Reviewer 4 Report

My comments have been thoroughly addressed. It is acceptable in the present form.